# Diffusion Facial Forgery Detection

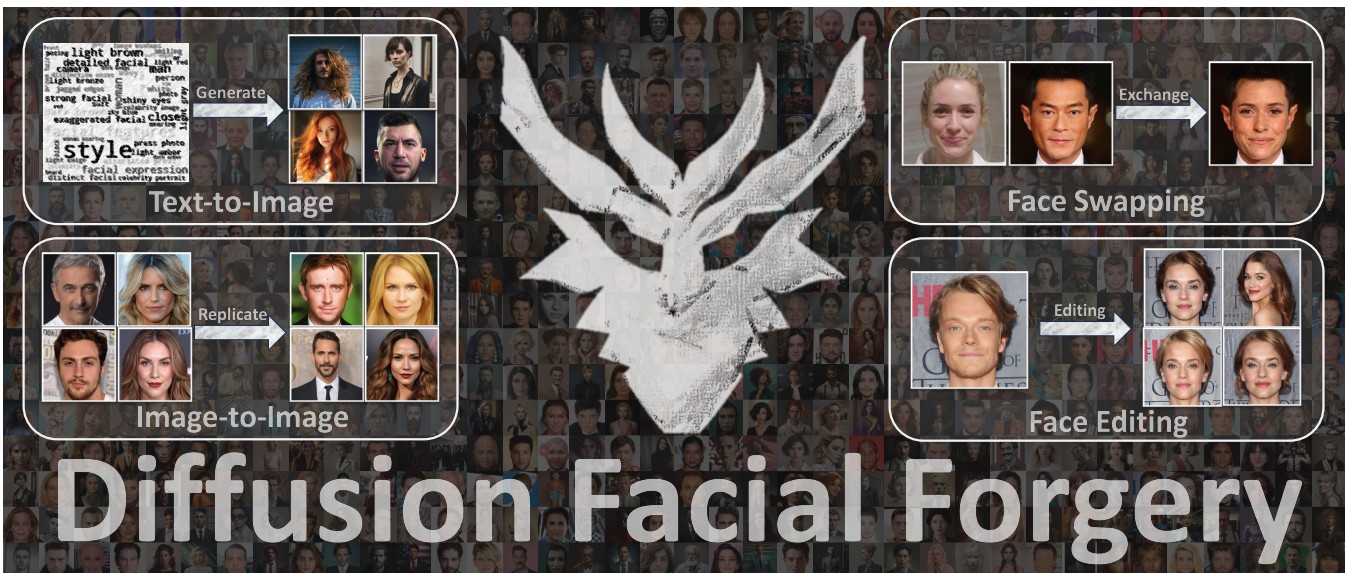

**Figure 1: DiFF – a diffusion-generated facial forgery dataset encompassing over half a million images. The dataset contains manipulated images created by thirteen state-of-the-art methods under four distinct conditions.**

## ABSTRACT

Detecting diffusion-generated images has recently grown into an emerging research area. Existing diffusion-based datasets predominantly focus on general image generation. However, facial forgeries, which pose severe social risks, have remained less explored thus far. To address this gap, this paper introduces DiFF, a comprehensive dataset dedicated to face-focused diffusion-generated images. DiFF comprises over 500,000 images that are synthesized using thirteen distinct generation methods under four conditions. In particular, this dataset utilizes 30,000 carefully collected textual and visual prompts, ensuring the synthesis of images with both high fidelity and semantic consistency. We conduct extensive experiments on the DiFF dataset via human subject tests and several representative forgery detection methods. The results demonstrate that the binary detection accuracies of both human observers and automated detectors often fall below 30%, revealing insights on the challenges in detecting diffusion-generated facial forgeries. Moreover, our experiments demonstrate that DiFF, compared to previous facial forgery datasets, contains a more diverse and realistic range of forgeries, showcasing its potential to aid in the development of more generalized detectors. Finally, we propose an edge graph regularization approach to effectively enhance the generalization capability of existing detectors.

## CCS CONCEPTS

• **Information systems** → **Multimedia databases**; • **Computing methodologies** → **Computer vision**.

## KEYWORDS

Diffusion-based Generation, Deepfake Detection, Facial Forgery Detection

**ACM Reference Format:**
Anonymous Author(s). 2024. Diffusion Facial Forgery Detection. In *Proceedings of ACM Multimedia (ACM MM)*. ACM, New York, NY, USA, 10 pages. https://doi.org/10.1145/nnnnnnn.nnnnnnn

## 1 INTRODUCTION

Conditional Diffusion Models (CDMs) have achieved impressive results in the field of image generation [3, 49]. Utilizing simple inputs, such as natural language prompts, CDMs can generate images with a high degree of semantic consistency [14, 24, 71]. However, the precise control over the generation process offered by CDMs has also raised concerns regarding security and privacy. For instance, malicious attackers can mass-produce counterfeit images of victims at a minimal cost, thus engendering negative social impacts.

Table 1: Comparison of DiFF and mainstream diffusion datasets. Existing diffusion datasets primarily focus on general arts and utilize limited conditional input. For generation conditions – T2I: Text-to-Image, I2I: Image-to-Image, FS: Face Swapping, FE: Face Editing. Pertaining to the Real Images column, *Source* represents that whether there is a real image collection process.

| Dataset | Venue | Type | #Synthetic Images | #Diffusion Methods | T2I | I2I | FS | FE | Source | Labels | Prompts |
|---|---|---|---|---|---|---|---|---|---|---|---|
| Stöckl *et al.* [56] | Arxiv'22 | General | 260K | 1 | ✓ | × | × | × | ✓ | × | Nouns of WordNet |
| De-Fake [52] | Arxiv'22 | General | 40K | 2 | ✓ | ✓ | × | × | ✓ | ✓ | Captions of the image dataset |
| Ricker *et al.* [45] | Arxiv'22 | General | 70K | 7 | × | × | × | × | × | × | Unconditional generation |
| TEdBench [25] | CVPR'23 | General | 0.1K | 1 | × | × | × | ✓ | ✓ | ✓ | 100 handwritten prompts for editing |
| DiffusionForensics [61] | ICCV'23 | General | 80K | 8 | ✓ | × | × | × | ✓ | × | 1 pre-defined template |
| DMDetection [9] | ICASSP'23 | General | 200K | 3 | ✓ | × | × | × | ✓ | ✓ | Captions of the image dataset |
| GenImage [75] | NeurIPS'23 | General | 1,300K | 5 | ✓ | × | × | × | ✓ | × | 1 pre-defined template |
| GFW [5] | Arxiv'22 | Facial | 15K | 3 | ✓ | × | × | × | × | × | Captions of the image dataset |
| Mundra *et al.* [38] | CVPRW'23 | Facial | 1.5K | 1 | ✓ | × | × | × | × | × | 10 pre-defined templates |
| DiFF (Ours) | – | Facial | 500K | 13 | ✓ | ✓ | ✓ | ✓ | ✓ | ✓ | 30K+ filtered high-quality prompts |

To address this problem, recent efforts have been made to collect datasets containing diffusion-generated images, wherein distribution differences [61] or amplitude variations [9] offers important cues for detection. Nevertheless, as shown in Table 1, these datasets are inadequate when applied to detect facial forgeries, which pose more significant threats than generic fake artifacts. Specifically, most large-scale diffusion-based datasets prioritize generic images [9, 25, 45, 52, 56, 61], like bedrooms and kitchens [68]. Although some face-related datasets have been introduced [5, 38], they all yet suffer from their small scales (*e.g.*, only 1.5K facial images in [38]). Moreover, these facial images are typically collected under restricted conditions with a narrow range of prompts, lacking comprehensive annotations as well. As a result, training a detector with generalizability on these datasets remains less viable.

This paper fills the gap by introducing the Diffusion Facial Forgery dataset, dubbed DiFF. There are three notable merits that make our dataset distinct from existing ones. i) To the best of our knowledge, our DiFF is the first comprehensive dataset that exclusively focuses on diffusion-generated facial forgery. It contains more than 500,000 facial forgery images, a scale that significantly surpasses previous facial datasets (as shown in Table 1). ii) DiFF is curated using a rich variety of diffusion methods and prompts. Specifically, it encompasses thirteen state-of-the-art diffusion techniques across four different conditions, including Text-to-Image, Image-to-Image, Face Swapping, and Face Editing. These methods are applied to generate high-quality images using over 20,000 carefully collected textual and 10,000 visual prompts, derived from 1,070 selected identities. iii) It is worth noting that each forged image in DiFF is meticulously annotated with the forgery method employed and the corresponding prompt.

We conduct in-depth human studies and extensive experiments upon the DiFF dataset. The results show that our DiFF includes forged images with more diversity and better reality than previous mainstream diffusion datasets, achieving over a 30% improvement in Fréchet Inception Distance (FID). Moreover, experiments with several deepfake and diffusion detectors [42, 48, 57, 61] reveal that existing detectors exhibit limited reliability in detecting diffusion-synthesized facial forgeries. For instance, the Xception model [48], originally designed for deepfake detection, achieves an Area Under

the receiver operating characteristic Curve (AUC) of only 60% on DiFF (versus 99% on conventional deepfake datasets). To overcome this issue, we propose a new regularization approach that leverages the edge graph of images to discern high-level facial features, thereby enhancing the generalizability of models. Our approach can be seamlessly integrated into existing detectors, achieving an average of 10% AUC improvements when applied to four popular detectors. This approach establishes an effective benchmark for the task of diffusion facial forgery detection.

The contributions of this paper are three-fold:

- We construct a diffusion-based facial forgery dataset with more than half a million images. To the best of our knowledge, this is the first large-scale dataset that focuses on high-quality diffusion-synthesized faces[1].
- We conduct extensive experiments on this dataset, which demonstrate that DiFF contains a richer and more realistic collection of synthetic images compared to previous diffusion datasets.
- We devise an approach based on edge graphs to identify the manipulated faces. Our approach can be seamlessly integrated into existing detection models, enhancing their detection ability and establishing comprehensive benchmarks for diffusion-generated face forgery detection.

## 2 RELATED WORK

### 2.1 Image Generation with Diffusion Models

Following the paradigm of introducing and then removing small perturbations from original images, diffusion models demonstrate the capability to generate high-quality images from white noise [54]. Early methods require no supervision signals and often perform unconditionally. For instance, Ho *et al.* [18] proposed a reverse learning process by estimating the noise in the image at each step. Subsequently, researchers have explored several optimization directions, including backbone architectures [4, 11, 49], sampling strategies [33, 39, 66], and adaptation for downstream tasks [1, 28, 70]. For example, Sinha *et al.* [53] proposed mapping latent representations to images using a diffusion decoding model. Song *et al.* [55]

---

[1]The dataset will be released upon the acceptance of this paper.

employed a non-Markovian forward process to construct denoising diffusion implicit models, resulting in a faster sampling procedure.

In contrast to the above unconditional approaches, recent diffusion models have shifted their focus toward conditional image synthesis [7, 22, 46, 50, 67, 69]. These conditions rely on various source signals, including class labels, textual prompts, and visual information, which generally describe specific image attributes. For instance, Cascaded Diffusion Models [19] initially generate low-resolution images from class labels and then employ subsequent models to increase resolutions. Furthermore, to achieve more detailed control, Text-to-Image Synthesis, which combines visual concepts and natural language, has emerged as one of the most notable approaches in diffusion models. These studies, exemplified by Stable Diffusion [41, 47], DALL-E [44], and Imagen [51], aim to align different modalities through pre-trained vision language models such as CLIP [43]. Additionally, some approaches leverage images as conditional inputs. Zhao *et al.* [72] utilized an energy-based function trained on both the source and target domains to generate images that preserve domain-agnostic characteristics. Lugmayr *et al.* [35] proposed an inpainting method that is agnostic to mask forms, altering reverse diffusion iterations by sampling unmasked regions from provided images.

## 2.2 Synthetic Image Detection

Detecting generated images has long been a popular research focus in computer vision. Earlier methods concentrate on the detection of specific types of forgeries, such as splicing [23], copy-move [34], or inpainting [30]. Thereafter, deep learning-based approaches have been applied to identify high-quality forgeries generated by GANs or diffusion models [59]. For instance, Frank *et al.* [13] proposed using frequency-domain features to detect forged images, as GAN models inevitably introduce artifacts during up-sampling. Guo *et al.* [15] presented a hierarchical fine-grained model to learn both comprehensive features and the inherent hierarchical nature of different forgery attributes.

Many recent studies have been dedicated to facial forgery detection [6, 60, 62]. Thus far, the majority of them have focused on the detection of swapped faces generated by VAE or GAN, *i.e.*, deepfakes [32]. For example, Masi *et al.* [36] introduced a two-branch network to extract optical and frequency artifacts separately. Real-Forensics [16] leverages visual and auditory correspondences in real videos to improve detection performance. Huang *et al.* [21] derived explicit and implicit embeddings using face recognition models, and the distance between these features serves as the foundation for distinguishing real from fake faces. With the rapid development of diffusion models, the risk posed by using them to generate counterfeit faces is gradually increasing [27]. However, research on the detection of diffusion-generated faces remains relatively unexplored. Although preliminary efforts have contributed to the detection of diffusion-generated outputs [9, 45, 61], they often lack generalizability and do not specifically focus on the detection of facial forgery.

## 3 DATASET CONSTRUCTION

In this work, our objective is to construct a high-quality dataset for diffusion-based facial forgery. The dataset is composed of three

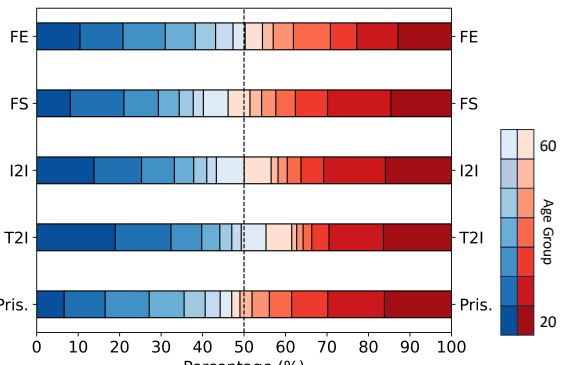

**Figure 2: Gender and age group distribution of pristine and forgery subsets. Within each subset, percentages for different ages (ranging from 20 to 60) are calculated separately for males (blue bars) and females (red bars).**

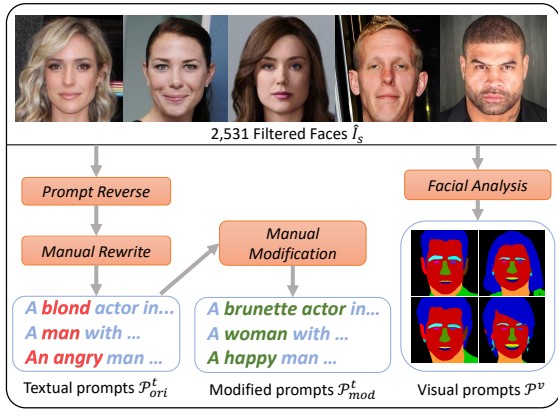

**Figure 3: Pipeline of prompts construction and modification.**

essential components: pristine images, prompts, and forged images. The pristine images constitute the *real (original)* instances of our dataset. Derived from these pristine images, the prompts serve as textual descriptions or visual cues that guide the diffusion model in generating forged images. We maintain a high degree of semantic consistency between pristine and forged images via these prompts.

## 3.1 Pristine Image Collection

Our pristine images are sourced from a pool of celebrity identities. Specifically, we manually select 1,070 celebrities from established celebrity datasets such as VoxCeleb2 and CelebA [2, 8, 29]. Figure 2 illustrates that we have ensured a balanced gender distribution and diverse age groups among these identities. In particular, the age distribution of the selected celebrities ranges from 20 to 60 across different subsets. Subsequently, we curate approximately 20 images per identity from public resources, resulting in a pristine collection, denoted as $\mathcal{I}_{pri}$, which encompasses a total of 23,661 images.

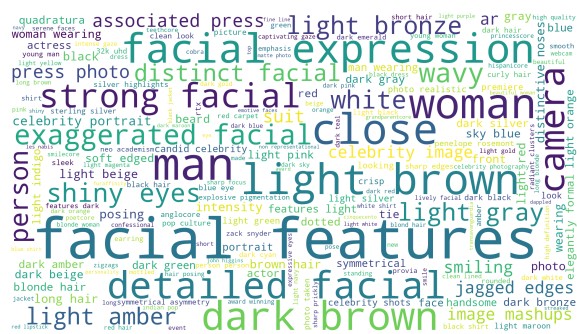

**Figure 4: Word cloud of the top 200 most frequent and content words in $\mathcal{P}^t_{ori}$. Each word size is proportioned to its frequency.**

## 3.2 Prompts Construction and Modification

Prior studies have demonstrated a positive correlation between the quality of conditional inputs and that of diffusion-generated images [40]. As a result, diverse and precise prompts are particularly useful for generating high-quality images in CDMs. Figure 3 illustrates that the construction of our dataset includes three categories of prompts: original textual prompts $\mathcal{P}^t_{ori}$, modified textual prompts $\mathcal{P}^t_{mod}$, and visual prompts $\mathcal{P}^v$. These prompts serve as conditions to guide the sampling process of diffusion models. The construction processes of these prompts are detailed below.

- **Original textual prompts $\mathcal{P}^t_{ori}$.** We generate diverse and natural textual prompts via a semi-automated approach. Initially, we curate a set of 2,531 high-quality images $\hat{I}_s \subset I_{pri}$ by selecting the clearest images of the frontal face for each identity. These images are then converted into textual descriptions using prompt inversion tools [10, 37]. These descriptions are reviewed and rewritten by experts to remove irrelevant terms and improve clarity. Consequently, we obtain 10,084 polished prompts, and some frequent words are shown in Figure 4.
- **Modified textual prompts $\mathcal{P}^t_{mod}$.** To broaden the diversity of prompts and enable the generation of images with specific modifications, $\mathcal{P}^t_{mod}$ involves alterations in key attributes of $\mathcal{P}^t_{ori}$. In particular, we randomly modify the salient words that describe identities in $\mathcal{P}^t_{ori}$, such as gender, hair color, or facial expression. For instance, we transform a prompt like 'A man with an emotive face' into 'A woman with an emotive face.'
- **Visual prompts $\mathcal{P}^v$.** These prompts comprise comprehensive facial features - such as sketches, landmarks, and segmentations - extracted from each image in $\hat{I}_s$. These features are applied for diffusion models conditioned on visual cues, which is particularly useful in tasks like face editing.

## 3.3 Facial Forgery Generation

As illustrated in Figure 5, we categorize existing CDMs into four main subsets based on their input types: Text-to-Image (T2I), which operates on textual prompts; Image-to-Image (I2I) and Face Swapping (FS), both of which utilize visual inputs; and Face Editing (FE), which incorporates a combination of text and visual conditions.

**Table 2: Detailed statistics of DiFF. We employ thirteen different methods to synthesize high-quality results based on 2.5K pristine images and their corresponding 20k textual and 10k visual prompts.**

| Subset | Method | #Images | Remarks |
|--------|--------|---------|---------|
| **T2I** | Midjourney [37] | 40,684 | Web Service |
| | SDXL [41] | 40,336 | Enhanced Stable Diffusion |
| | FreeDoM_T [69] | 18,207 | ICCV'23 |
| | HPS [64] | 36,464 | ICCV'23 |
| **I2I** | LoRA [20] | 42,800 | LoRA adaption for diffusion |
| | DeamBooth [50] | 40,526 | CVPR'23 |
| | SDXL Refiner [41] | 40,336 | Refine module for SDXL |
| | FreeDoM_I [69] | 43,593 | ICCV'23 |
| **FS** | DiffFace [26] | 55,693 | First diffusion-based FS work |
| | DCFace [27] | 44,721 | CVPR'23 |
| **FE** | Imagic [25] | 40,508 | CVPR'23 |
| | CoDiff [22] | 48,672 | CVPR'23 |
| | CycleDiff [63] | 44,926 | ICCV'23 |
| | **Total** | 537,466 | - |

Figure 5a demonstrates that T2I methods receive textual prompts (*e.g.*, 'A man in uniform') and synthesize images that align with the inputs' semantic content [47]. In contrast, models processing visual input are further divided into I2I and FS categories based on their manipulation processes. Specifically, I2I, as illustrated in Figure 5b, pertains to methods that replicate a single identity. On the other hand, FS models simultaneously handle two identities and perform identity swaps as presented in Figure 5c. Lastly, Figure 5d highlights that FE models utilize multi-modal inputs to modify facial attributes, such as expressions or lip movements, while preserving other attributes. These four subsets achieve comprehensive coverage of the conditions under which existing diffusion models operate. Moreover, to ensure the diversity of generated faces, we utilize multiple cutting-edge techniques within each category. A detailed introduction to these methods is as follows:

**Text-to-Image.** We employ four state-of-the-art methods - *Midjourney* [37], *Stable Diffusion XL (SDXL)* [41], *FreeDoM_T* [69], and *HPS* [64] - for this subset. The first two are the most influential web services for which we employ official APIs. The latter two are recently released T2I models, and we apply their pre-trained models. These models are guided by textual prompts $\mathcal{P}^t_{ori}$.

**Image-to-Image.** We apply four methods in this context: *Low-Rank Adaption (LoRA)* [20], *DreamBooth* [50], *SDXL Refiner* [41], and *FreeDoM_I*. Among these approaches, the former two require fine-tuning of diffusion models to capture specific facial features. We employ $I_{pri}$ to train these two models. SDXL Refiner optimizes results from SDXL, whereas FreeDoM_I substitutes the textual encoder in FreeDoM_T with a visual encoder to reconstruct faces.

**Face Swapping.** In this subset, we implement *DiffFace* [26] and *DCFace* [27] for the face swapping task. For each image in $\hat{I}_s$, we randomly choose ten targets from other identities to perform face swaps. In particular, to prevent information leakage, we divide the 1,070 identities into disjoint training, validation, and testing sets in an 8:1:1 ratio.

**Face Editing.** This subset involves three approaches. In particular, the modified textual prompt set $\mathcal{P}^t_{mod}$ and the pristine image set $\hat{I}_s$ are both used in *Imagic* [25] and *Cycle Diffusion (CycleDiff)* [63]

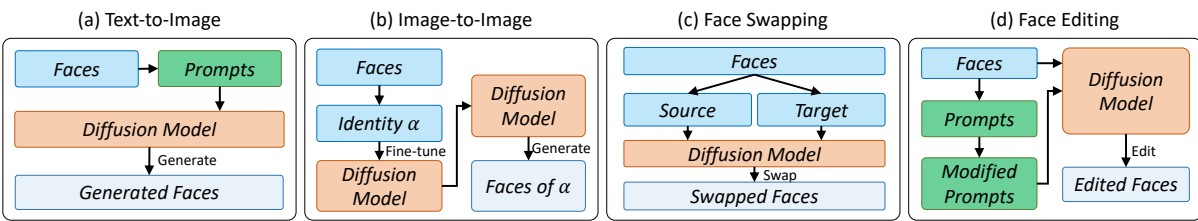

**Figure 5: Facial forgery generation under four conditions.**

**Table 3: Human performance (%) on DiFF.**

| Text-to-Image | | Image-to-Image | | Face Swapping | | Face Editing | |
|---|---|---|---|---|---|---|---|
| Method | ACC | Method | ACC | Method | ACC | Method | ACC |
| Midjourney | 65.32 | SDXL Refiner | 71.85 | DiffFace | 36.33 | Imagic | 68.17 |
| SDXL | 72.11 | LoRA | 33.33 | DCFace | 66.67 | CoDiff | 27.78 |
| FreeDoM_T | 25.47 | DreamBooth | 76.65 | | | CycleDiff | 40.65 |
| HPS | 75.68 | FreeDoM_I | 56.67 | | | | |

to edit faces. Moreover, we use visual prompts $\mathcal{P}^v$ to guide the training of *Collaborative Diffusion (CoDiff)* [22].

In summary, we show the statistics pertaining to the images generated by the aforementioned methods in Table 2. As can be observed, the total number of generated images is over 500K from thirteen diffusion methods.

## 4 DATASET EVALUATION

Following the methodologies in deepfake detection [48], we cast the detection of diffusion-generated facial forgeries as a binary classification task.

### 4.1 Human Evaluation

We conducted comprehensive human studies involving 70 participants. In this study, participants are instructed to classify the authenticity of randomly selected images that are generated from varied approaches. The image selection followed a 50:50 split between pristine and fake images, with each identity appearing only once to prevent bias. Each participant is required to carefully examine 200 images, yielding 14,000 human results in total.

Table 3 presents the accuracy (ACC) of this experiment across all forgery methods under four conditions. One can observe that human observers struggle to distinguish the vast majority of forgery methods, as accuracy falls below the chance level (50%). For instance, participants achieved an accuracy of 27.78% when identifying images generated by CoDiff. Among the four conditions, FE poses the most challenge for human observers. This result can be attributed to the fact that this subset involves manipulations of modifying a single real image, which allows for a more faithful reproduction of original features.

### 4.2 Comparison with Existing Datasets

*4.2.1 Statistics analysis.* We presented the Fréchet Inception Distance (FID) and Peak Signal-to-Noise Ratio (PSNR) metrics in Table 4. The reported results reveal notable improvements in DiFF. For instance, the FID of DiFF shows an approximate improvement of 30% and 20% over the previous diffusion datasets GFW and DiffusionForensics (DFor), respectively, which suggesting that images in our dataset bear a closer resemblance to reality.

**Table 4: FID and PSNR comparison across various datasets.**

| Dataset | FF++ [48] | ForgeryNet [17] | DFor [61] | GFW [5] | DiFF |
|---|---|---|---|---|---|
| FID ↓ | 33.87 | 36.94 | 31.79 | 39.35 | **25.75** |
| PSNR ↑ | 18.47 | 18.98 | 19.17 | 19.14 | **19.95** |

**Table 5: AUC (%) of detectors trained and tested on same datasets.**

| Method | Dataset | | |
|---|---|---|---|
| | FF++ [48] | GFW [5] | DiFF |
| Xception | 98.12 | 99.72 | 93.87 |
| F³-Net | 98.89 | 99.17 | 98.47 |
| EfficientNet | 98.51 | 97.58 | 94.34 |
| DIRE | 99.43 | 99.59 | 96.35 |

**Table 6: AUC (%) of detectors trained on different datasets.**

| Method | Train Set | Test Set | | | | | |
|---|---|---|---|---|---|---|---|
| | | FF++ [48] | DFor [61] | GFW [5] | DiFF | DFDC [12] | ForgeryNet [17] |
| Xception | FF++ | - | 40.65 | 43.42 | 65.96 | 63.97 | 50.56 |
| | DFor | 55.21 | - | 52.30 | 75.67 | 56.35 | 38.06 |
| | GFW | 53.37 | 45.81 | - | 74.87 | 51.43 | 62.75 |
| | DiFF | **65.33** | **55.30** | **63.50** | - | **67.10** | **65.78** |

*4.2.2 Trained on DiFF and other face forgery datasets.* Beyond the observed benefits in terms of FID and PSNR, Table 5 indicates that the scores of Area Under the receiver operating characteristic Curve (AUC) of DiFF are relatively lower, highlighting the dataset's higher complexity. This can be attributed to the extensive diversity of conditions and image synthesis methods in DiFF, both of which are three times greater than those in GFW.

Moreover, we utilized FF++ (vanilla deepfake dataset, HQ version), DFor (diffusion-generated general forgery dataset), GFW (diffusion-generated facial forgery dataset), and our DiFF as training datasets to compare their generalization capabilities. Additionally, we utilized two widely acknowledged deepfake datasets, DFDC [12] and ForgeryNet [17], for further evaluation. From Table 6, it can be seen that the detector trained on DiFF exhibits superior generalization capabilities. It is worth noting that detectors trained on other datasets achieve high accuracy when tested on DiFF. This may be attributed to DiFF's inclusion of a diverse array of image types, effectively encompassing a wide spectrum of distributions.

### 4.3 Detection Results of Existing Methods

For this experiment, we split our DiFF dataset into training, validation, and testing sets with a 8:1:1 ratio. We tuned the hyperparameters with the validation set, and reported results on the testing set.

**Table 7: AUC (%) of detectors. Each detector is trained on the deepfake dataset (FF++) and the diffusion-generated general forgery dataset (DFor) separately, and tested on subsets of the DiFF dataset. †: models for deepfake detection. ‡: models for general diffusion detection.**

| Deepfake | | Test Subset | | | | |
|---|---|---|---|---|---|---|
| Method | Train Set | FF++ | T2I | I2I | FS | FE |
| Xception† [48] | FF++ [48] | 98.12 | 62.43 | 56.83 | 85.97 | 58.64 |
| F³-Net† [42] | | 98.89 | 66.87 | 67.64 | 81.01 | 60.60 |
| EfficientNet† [57] | | 98.51 | 74.12 | 57.27 | 82.11 | 57.20 |
| DIRE‡ [61] | | 99.43 | 44.22 | 64.64 | 84.98 | 57.72 |
| General Diffusion | | Test Subset | | | | |
| Method | Train Set | DFor | T2I | I2I | FS | FE |
| Xception† [48] | DFor [61] | 99.98 | 20.52 | 30.92 | 69.42 | 37.89 |
| F³-Net† [42] | | 99.99 | 43.88 | 60.58 | 52.39 | 47.06 |
| EfficientNet† [57] | | 98.99 | 27.23 | 44.79 | 61.25 | 30.86 |
| DIRE‡ [61] | | 98.80 | 36.37 | 34.83 | 36.28 | 39.92 |

*4.3.1 Cross-domain Detection.* Following prior studies on forgery detection [61], we adopted a cross-domain testing methodology to explore the challenges of facial forgery detection. This involves evaluating models that have performed well in related detection domains. Initially, these models are trained on benchmark datasets tailored to their respective tasks. We then evaluated their performance on DiFF. Three widely recognized deepfake detection models are utilized: Xception [48], F³-Net [42], and EfficientNet [57]. Moreover, we included DIRE [61], a state-of-the-art detector for general diffusion-generated images, for this experiment. These models are trained on the FF++ dataset [48] and the DiffusionForensics dataset [61], respectively.

Table 7 displays the AUC scores for these detectors. From this table, we can observe that these detectors encounter a significant drop in performance upon domain transfer. For example, DIRE exhibits an AUC drop of over 60%. This sharp degradation indicates the inherent challenge of detecting diffusion-based facial forgeries and suggests the considerable obstacles that pre-trained detectors face when applied to this new task.

*4.3.2 In-domain Detection.* Given that existing deepfake and general diffusion detectors cannot be seamlessly transferred to detect diffusion facial forgery, one may question the efficacy of re-training these detectors on DiFF. Therefore, we conducted experiments with an in-domain setting. Similar to previous evaluation protocols for the detection of deepfake and general diffusion forgery [31, 61], detectors are trained on a single subset of DiFF, followed by the test on the remaining ones.

**Detection on re-training detectors.** We presented the re-training results in Figure 6. It can be observed that detectors perform satisfactorily when trained and tested on the same subset. However, when transferred to different subsets, they exhibit varying degrees of performance degradation. The most significant drop reaches up to 80% (*e.g.*, Xception, trained on FS and tested on FE). This significant drop highlights the challenges in developing a facial forgery detector that effectively generalizes across various conditions.

It is worth noting that detectors trained on the T2I and I2I subsets, which both rely on classical diffusion processes, demonstrate a higher degree of similarity in performance. This is evident from

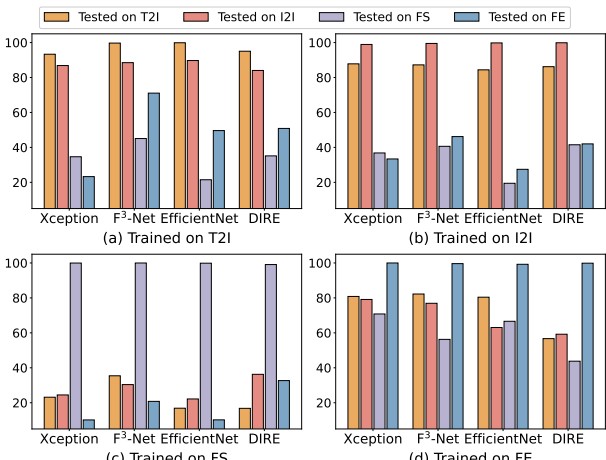

**Figure 6: AUC (%) comparison among re-trained detectors.**

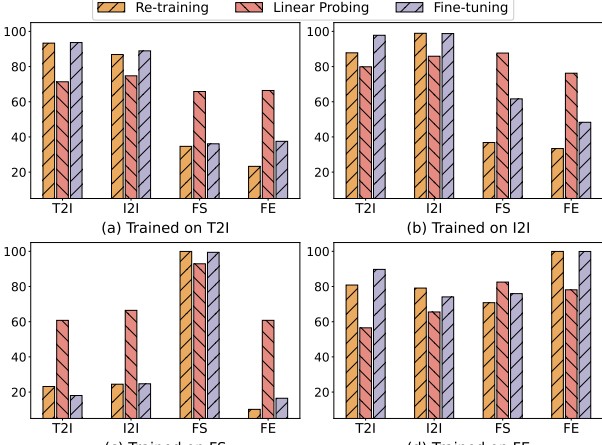

**Figure 7: AUC (%) of Xception with different training strategies.**

mutual benefits observed in the first two subplots of Figure 6. FE-trained detectors show better generalization capability than those trained in other subsets. This may be attributed to the FE subset's utilization of multi-modal inputs, leading to a wider diversity of images, thereby enabling detectors trained on the FE subset to capture more diffusion artifacts.

**Detection on linear probing detectors.** We introduced the strategy of linear probing as an alternative to the full re-training approach. Specifically, we used Xception pre-trained on the FF++ dataset as described in Section 4.3.1 and optimized its last linear layer to align with the data distribution of DiFF. The results are presented in Figure 7.

One can observe that models using the linear probing strategy significantly outperform the re-training ones in detecting FS and FE forgeries. For instance, when trained on the I2I subset, the linear probing model for detecting FS and FE forgeries surpass the re-training models by 50% and 40%, respectively. A critical reason is that the pre-training dataset, *i.e.*, FF++, encompasses a large number of GAN-based manipulated faces. This diversity enables

**Table 8: AUC (%) comparison among re-trained detectors with different post-processing methods. Each row represents the average performance when tested on all four DiFF subsets. *None*: no processing methods. *GN*: Gaussian Noise; *GB*: Gaussian Blur; *MB*: Median Blur; *JPEG*: JPEG Compression.**

| Method | Train Subset | Processing Method | | | | |
|---|---|---|---|---|---|---|
| | | None | GN | GB | MB | JPEG |
| Xception | T2I | 59.52 | 47.65 | 15.02 | 56.59 | 58.69 |
| F³-Net | | 76.08 | 48.04 | 74.67 | 71.68 | 74.61 |
| EfficientNet | | 67.69 | 40.09 | 53.62 | 65.35 | 54.98 |
| DIRE | | 66.28 | 34.07 | 32.78 | 41.36 | 40.99 |
| Xception | I2I | 66.74 | 19.70 | 54.09 | 58.07 | 63.66 |
| F³-Net | | 68.39 | 21.38 | 58.77 | 66.17 | 63.61 |
| EfficientNet | | 57.78 | 27.76 | 54.75 | 52.39 | 51.01 |
| DIRE | | 67.40 | 35.69 | 26.63 | 65.19 | 65.67 |
| Xception | FS | 39.44 | 35.40 | 34.82 | 38.58 | 37.73 |
| F³-Net | | 46.64 | 44.44 | 37.91 | 46.39 | 42.10 |
| EfficientNet | | 37.29 | 36.74 | 23.82 | 36.12 | 35.13 |
| DIRE | | 46.03 | 25.36 | 34.00 | 28.15 | 32.11 |
| Xception | FE | 82.69 | 39.69 | 24.15 | 79.35 | 81.19 |
| F³-Net | | 78.84 | 50.17 | 25.51 | 38.76 | 70.31 |
| EfficientNet | | 77.33 | 51.95 | 39.65 | 71.10 | 71.14 |
| DIRE | | 64.89 | 35.42 | 55.59 | 60.08 | 53.08 |

linear probing models to effectively identify face-swapping and face-editing images. However, it is worth noting that linear probing models show inferior results when trained and tested on the same subset (*e.g.*, both trained and tested on T2I), suggesting insufficient capacity of this strategy.

**Detection on fine-tuning detectors.** In contrast to the linear probing strategy, which updates only the final layer, the fine-tuning approach optimizes all the model parameters. For this experiment, we reduce the learning rate of models for stable training. Figure 7 illustrates that fine-tuning models demonstrate superior performance compared to the re-training ones. For example, fine-tuning models achieve higher AUCs in the detection of FS and FE forgeries, regardless of the training subset used. This can also be attributed to the pre-training on FF++. However, compared to linear probing models, the generalizability of the fine-tuning approaches is somewhat limited. This may be due to a significant discrepancy between diffusion-generated facial forgeries and GAN-based manipulated faces. Such a domain gap could lead to catastrophic forgetting.

**Detection with post-processing methods.** We evaluated re-training detectors under various image quality settings by applying several post-processing techniques. Following previous settings [74], we processed real and forged images with Gaussian Noise (GN), Gaussian Blur (GB), Median Blur (MB), and JPEG Compression (JPEG). Table 8 reveals that, in most scenarios, applying post-processing methods leads to a degradation in the detection performance. For instance, the use of GB results in a 40% reduction in the AUC for Xception when trained on the T2I subset.

## 5 EDGE GRAPH REGULARIZATION

### 5.1 Motivation

Compared to real human faces captured by cameras, generated faces are more likely to evoke anomalies such as frequency transitions or brightness fluctuations [45]. In particular, we extract the edge graphs of pristine and forged images with the Sobel operator [58]. Figure 8 illustrates that the edge graphs of the pristine

images are significantly different from those of the synthesized images. Specifically, edge graphs extracted from pristine images often capture intricate facial details, such as fine wrinkles around the cheeks. In contrast, the synthesized images lack these subtle contours and are of considerable contrast.

One naive approach to exploit the discriminative capability of edge graphs for facial forgery detection is to train a binary classifier directly. However, our experiments indicate that this approach is less favorable, as demonstrated in Section 5.3.3. Instead, we propose an Edge Graph Regularization (EGR) method, which enhances the discriminative ability of detectors by incorporating edge graphs into the processing of original images.

### 5.2 Methodology

Vanilla deepfake and general diffusion detecors entails fitting the distribution of a specific dataset to discriminate between pristine and forged images. Let $\mathcal{S} = \{(\mathbf{I}_i, y_i)\}_{i=1}^{n}$ be the dataset, where $\mathbf{I}_i$ is the $i$-th image with respect to the target label $y_i$. For each parameter set $\theta \in \Theta$, wherein $\Theta$ represents the continuous parameter space, the empirical risk during training is formulated as follows:

$$\hat{R}_{\mathcal{S}}(\theta) := \frac{1}{n} \sum_{i=1}^{n} \ell\left(\theta, \mathbf{I}_i, y_i\right), \tag{1}$$

where $\ell(\cdot)$ is the loss function such that,

$$\ell(\theta, \mathbf{I}_i, y_i) = -(y_i \log(\hat{y}_i) + (1 - y_i) \log(1 - \hat{y}_i)), \tag{2}$$

where $\hat{y}_i$ is the score from the predictive function $f_\theta : \mathbf{I}_i \rightarrow [0, 1]$ associated with $\theta$. However, such training approaches are highly susceptible to overfitting [31, 48, 65]. Therefore, many endeavors have been made to improve generalizability using additional features [21, 73]. In light of these studies, our method employs a novel regularization method, which incorporates edge graphs as a regularization term into the empirical risk. This strategy encourages the model to simultaneously focus on the features of both the original and edge graphs, thereby mitigating overfitting. Specifically, we refine the empirical risk as follows:

$$R_{\mathcal{S}}(\theta) := \hat{R}_{\mathcal{S}}(\theta) + \lambda \frac{1}{n} \sum_{i=1}^{n} (\ell\left(\theta, \mathbf{E}_i, y_i\right)), \tag{3}$$

where $\mathbf{E}_i$ represents the edge graph of the $i$-th image, and $\lambda \in [0, 1]$ is a regularization parameter that calibrates the influence of edge graphs.

### 5.3 Evaluation of EGR

*5.3.1 Main results.* In Table 9, we compared the performance of baseline detectors and our method. Each model is trained on one forgery condition and subsequently evaluated on all four conditions. From the table, one can observe that our EGR method significantly improves the generalizability of the baseline detectors. It is worth noting that even when a model is trained and tested in the same subset, EGR still contributes to performance enhancement, such as improving Xception with 2.2% AUC on T2I.

*5.3.2 RoC Curves.* In Figure 9, we presented the ROC curves of EfficientNet trained on the FE subset and tested on the T2I subset, the full DiFF, and GFW. It can be observed that EGR significantly enhances the model's performance across various datasets, further

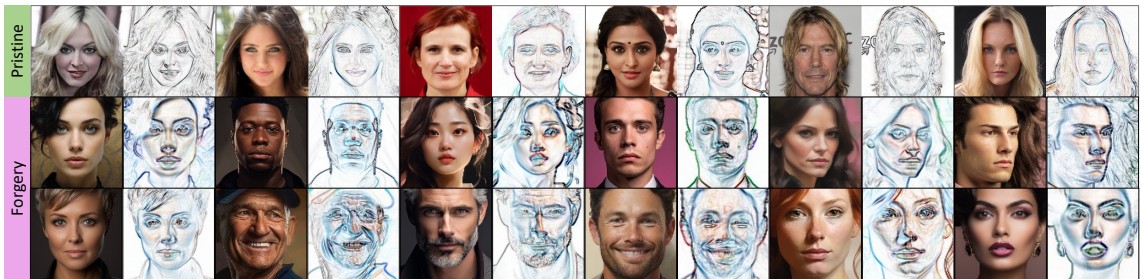

**Figure 8: Edge graphs of pristine images (first row) and non-cherry-picked forged facial images (last two rows).**

**Table 9: Model performance (%) with and without our EGR method. Each row represents the performance of the model trained on a specific subset and tested on all four DiFF subsets. Better results are highlighted in bold.**

| Method | | Train | Test Subset | | | |
|---|---|---|---|---|---|---|
| Backbone | +EGR | Subset | T2I | I2I | FS | FE |
| Xception | ✗ | | 93.32 | 86.85 | 34.65 | 23.28 |
| Xception | ✓ | | **95.57** | **89.48** | **43.74** | **55.50** |
| F³-Net | ✗ | | 99.60 | 88.50 | 45.07 | 71.06 |
| F³-Net | ✓ | T2I | **99.64** | **93.30** | **56.34** | **79.89** |
| EfficientNet | ✗ | | 99.89 | 89.72 | 21.49 | 49.63 |
| EfficientNet | ✓ | | **99.93** | **97.89** | **40.86** | **52.36** |
| DIRE | ✗ | | 95.04 | 84.07 | 35.15 | 50.86 |
| DIRE | ✓ | | **99.79** | **99.76** | **43.59** | **66.41** |
| Xception | ✗ | | 87.82 | 98.92 | 36.82 | 33.39 |
| Xception | ✓ | | **99.00** | **99.94** | **49.73** | **33.81** |
| F³-Net | ✗ | | 87.23 | 99.50 | 40.62 | 46.19 |
| F³-Net | ✓ | I2I | **96.85** | **99.70** | **48.69** | **47.66** |
| EfficientNet | ✗ | | 84.39 | 99.80 | 19.47 | 27.46 |
| EfficientNet | ✓ | | **99.77** | **99.99** | **56.69** | **61.04** |
| DIRE | ✗ | | 86.20 | 99.88 | 41.51 | 42.01 |
| DIRE | ✓ | | **97.65** | **99.99** | **51.84** | **58.68** |
| Xception | ✗ | | 23.17 | 24.47 | 99.95 | 10.17 |
| Xception | ✓ | | **67.41** | **55.92** | **99.98** | **46.01** |
| F³-Net | ✗ | | 35.43 | 30.39 | 99.98 | 20.79 |
| F³-Net | ✓ | FS | **63.51** | **63.75** | **99.99** | **31.14** |
| EfficientNet | ✗ | | 16.88 | 22.17 | 99.87 | 10.21 |
| EfficientNet | ✓ | | **64.16** | **67.92** | **99.99** | **22.01** |
| DIRE | ✗ | | 16.08 | 36.27 | 99.09 | 32.68 |
| DIRE | ✓ | | **66.21** | **70.91** | **99.99** | **35.45** |
| Xception | ✗ | | 80.84 | 79.12 | 70.81 | 99.95 |
| Xception | ✓ | | **94.15** | **84.04** | **73.09** | **99.99** |
| F³-Net | ✗ | | 82.32 | 76.92 | 56.27 | 99.60 |
| F³-Net | ✓ | FE | **97.91** | **93.46** | **79.33** | **99.61** |
| EfficientNet | ✗ | | 80.41 | 63.06 | 66.62 | 99.24 |
| EfficientNet | ✓ | | **96.50** | **89.97** | **73.28** | **99.99** |
| DIRE | ✗ | | 56.70 | 59.22 | 43.78 | 99.87 |
| DIRE | ✓ | | **81.40** | **76.40** | **74.23** | **99.99** |

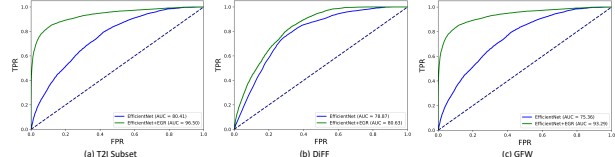

**Figure 9: ROC curves of EfficientNet trained on the FE subset.**

demonstrating the effectiveness of the EGR method in developing generalized diffusion detectors.

*5.3.3 Ablation study.* To evaluate the impact of the proposed EGR method, we conducted experiments using edge graphs as the only input. In other words, we removed the $\hat{R}_{\mathcal{S}}(\theta)$ in Equation (3), and

**Table 10: AUC (%) comparison of detectors with the removal of the regularization approaches. All models are trained on the T2I subset.**

| Method | Test Subset | | | |
|---|---|---|---|---|
| | T2I | I2I | FS | FE |
| Xception | **95.57** | **89.48** | **43.74** | **55.50** |
| *w/o* regu. | 95.54(-0.03) | 88.91(-0.57) | 43.31(-0.43) | 53.21(-2.29) |
| F³-Net | **99.64** | **93.30** | **56.34** | **79.89** |
| *w/o* regu. | 97.83(-1.81) | 93.02(-0.28) | 51.50(-4.84) | 64.80(-15.09) |
| EfficientNet | **99.93** | **97.89** | **40.86** | **52.36** |
| *w/o* regu. | 98.97(-0.96) | 96.34(-1.55) | 26.09(-14.77) | 49.82(-2.54) |
| DIRE | **99.79** | **99.76** | **43.59** | **66.41** |
| *w/o* regu. | 99.78(-0.01) | 99.70(-0.06) | 32.36(-11.23) | 61.61(-4.80) |

optimized the model with $\mathbf{E}_i$. The results of these tests are presented in Table 10. We can observe a significant decline in detector performance upon removing the regularization approach. For instance, in the FE subset, the AUC of F³-Net drops by 15%. The dominant reason is that relying solely on edge graphs overlooks vital information in original images, such as color and texture. On the other hand, incorporating the EGR enables the model to capture a more broad context, leading to better performance. In a nutshell, the combined utilization of edge graphs and color information achieves optimal results.

## 6 DISCUSSION AND CONCLUSION

We propose DiFF, a large-scale high-quality diffusion-generated facial forgery dataset, to address limitations of existing datasets that underestimate the risks associated with facial forgeries. Our dataset comprises over 500,000 facial images. Each image maintains high semantic consistency with its original counterpart, guided by diverse prompts. We conduct extensive experiments using DiFF and establish a facial forgery detection benchmark. Moreover, we design an edge graph regularization method that effectively improves detector generalization performance. In the future, we plan to further expand DiFF in terms of generative algorithms and conditions and explore new tasks based on DiFF, such as the traceability and retrieval of diffusion-generated images.

**Potential Ethical Considerations.** The pristine faces in our dataset are sourced from publicly accessible celebrity online videos. We have rigorously reviewed all prompts to ensure that they do not describe specific biometric details. Each generated image has been carefully examined to align with societal values. We will try our best to control the acquisition procedure of our DiFF to mitigate potential misuse.

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
