# OpenReview forum: "Diffusion Facial Forgery Detection"
_acmmm.org/ACMMM/2024/Conference — MM2024 Poster_

### Official Review · Reviewer_H5Rr · 2024-05-02

**Rating:** 4
**Confidence:** 4

**Summary:**

This paper introduces a comprehensive dataset dedicated to face-focused diffusion-generated images, i.e., DiFF, which using a variety of diffusion models and carefully collected textual and visual prompts to generate a more diverse and realistic range of forgeries. Extensive experiments are conducted on the proposed dataset, which demonstrate that DiFF contains a richer and more realistic collection of synthetic images compared to previous diffusion datasets. In addition, the author proposes an edge graph regularization approach to enhance the generalization performance for deepfake detection.

**Strengths:**

The structure of this paper is clear, and the content is easy to follow.
To construct diffusion-based facial forgery dataset, the authors utilize c a variety of diffusion models and carefully collected textual and visual prompts to generate more than half a million face forgery images.
Comprehensive experimental evaluation substantiates the proposed DiFF includes forged images with more diversity and better reality than previous mainstream diffusion datasets.

**Limitations:**

Since this article is concerned with the construction of deepfake's dataset, but the related work only describes the generative model (i.e., diffusion model) and the synthetic image detection, there is no systematic presentation of other deepfake's datasets.

The proposed edge graph regularization approach, which essentially uses the Sobel operator to extract high frequency information, is not novel. This operation of extracting high-frequency information for improving detection performance has actually been widely used in deepfake detection.

**Suitability:**

2

---

### Official Review · Reviewer_72NN · 2024-05-21

**Rating:** 4
**Confidence:** 3

**Summary:**

The paper introduces DiFF, a large-scale diffusion-based facial forgery dataset with over 500k images generated by 13 diffusion methods. These methods are categorized into four types: text-to-image (T2I), image-to-image (I2I), face swapping (FS), and face editing (FE). This dataset significantly enriches the current pool of diffusion-based data, which predominantly focuses on general images. Additionally, the paper proposes a technique for extracting edge graphs to serve as supplementary input alongside RGB images, enhancing the effectiveness of existing deepfake detection methods.

**Strengths:**

1. The paper provides a large-scale, diverse dataset of facial forgeries created using 13 different diffusion methods across four comprehensive categories.

2. The dataset includes rich annotations with prompts in addition to labels.

3. The dataset appears to be balanced in terms of demographics and manipulation methods.

4. The proposed edge graph regularization method is shown to be effective.

**Limitations:**

1. The paper claims that "this is the first large-scale dataset that focuses on high-quality diffusion-synthesized faces." However, Țânțaru et al. (WACV 2024) [i] and Chen et al. (pre-print) [ii] have introduced similar datasets.

2. The exclusive use of high-quality images of celebrities may not be ideal. Since deepfakes can vary in quality depending on the attacker's purposes, it would be better to also include images of varying quality and distributions. This could involve using images of everyday people taken with different cameras under various conditions.

3. The use of PSNR for quality measurement is questioned. The paper does not clarify how PSNR was applied, and it may not be an appropriate metric in this context.

4. The paper asserts that the DiFF dataset is challenging, but detectors trained on other datasets perform well on DiFF. It would be more accurate to state that DiFF is more diverse, leading to higher test accuracy on other datasets when detectors are trained on DiFF.

5. There is a discrepancy between the descriptions in section 5.3.3 and Table 10. Did the authors intend to use "only" rather than "w/o" regulation in Table 10?

**References**

[i] Țânțaru, Dragoș-Constantin, Elisabeta Oneață, and Dan Oneață. "Weakly-supervised deepfake localization in diffusion-generated images." In Proceedings of the IEEE/CVF Winter Conference on Applications of Computer Vision, pp. 6258-6268. 2024.

[ii] Chen, Zhongxi, Ke Sun, Ziyin Zhou, Xianming Lin, Xiaoshuai Sun, Liujuan Cao, and Rongrong Ji. "DiffusionFace: Towards a Comprehensive Dataset for Diffusion-Based Face Forgery Analysis." arXiv preprint arXiv:2403.18471 (2024).

**Suitability:**

3

---

### Official Review · Reviewer_PAYy · 2024-05-23

**Rating:** 4
**Confidence:** 4

**Summary:**

This paper collects a large synthetic data set containing 13 generative models.
This paper constructs a wealth of experiments and proposes a method to enhance generalization ability.

**Strengths:**

This paper constructs a diffusion-based facial forgery dataset with more than half a million images. This is a boost for the fake face detection community.
The proposed EGR approach is interesting.

**Limitations:**

1. Lack of comparative experiments on data sets based on GAN models.
2. For section 5.2, detailed explanation and formula proof are lacking.
3. The motivation of the proposed method is not clear enough to understand the role of edge maps in alleviating overfitting.
4. Use Sobel operator to capture the edge graphs, which is widely used in the field of image forensics and lacks innovation.
5. From the visualization results in Figure 8, it can be observed that the true and false images are very similar, and it is difficult to detect the difference.
6. Lack of 3to1 experiment Settings, such as training in three generation methods and testing in one generation method.
7. There is a lack of cross-generation experiments, such as training in DDPM architecture and testing in LDM architecture.
8. This dataset lacks the exploration of local face tampering localization tasks and the discussion of related papers, such as:
[1] On the detection of digital face manipulation. CVPR 2020
[2] Inconsistency-aware wavelet dual-branch network for face forgery detection. TBIOM 2021
[3] FakeLocator: Robust localization of GAN-based face manipulations. TIFS 2022
[4] Multi-spectral Class Center Network for Face Manipulation Detection and Localization. Arxiv 2023

**Suitability:**

2

---

### Meta-Review · Area_Chair_1neF · 2024-07-04

**Recommendation:** Accept (Poster)
**Confidence:** 5

**Metareview:**

The initial ratings of this paper are positive. After the rebuttal by authors, PAYy changed its rating to weak reject, due to the concerns about the comprehensiveness of the experiments. AC agrees that the comparisons and literature reviews contain certain flaws, but the proposed dataset has its value to the community, which is also agreed by all reviewers. Therefore, AC decides to accept this paper, but the authors should seriously revise the paper according to the review comments and the rebuttal.